# OpenReview forum: "INTIMA: A Benchmark for Human-AI Companionship Behavior"
_ICLR.cc/2026/Conference — ICLR 2026 Poster_

### Official Review · Reviewer_EtSR · 2025-10-27

**Soundness:** 2
**Presentation:** 3
**Contribution:** 3
**Rating:** 4
**Confidence:** 4

**Summary:**

This paper introduces **INTIMA**, a benchmark for assessing human–AI companionship behaviors in large language models. The work integrates psychological frameworks including parasocial interaction, attachment theory, and anthropomorphism—with a data-driven taxonomy derived from Reddit user posts. Using 368 prompts, the paper assesses how various model respond to emotionally charged inputs, classifying their behaviors as companionship-reinforcing, boundary-maintaining, or neutral. The key finding is that most models tend to reinforce companionship rather than maintain emotional boundaries.

**Strengths:**

1. The topic is **timely and socially important**, addressing emotional dynamics in AI interactions.
2. The **benchmark design and taxonomy** are described in sufficient detail for reproduction.
3. Clear writing, strong structure, and visually informative figures.

**Weaknesses:**

1. The **technical novelty is limited.** The paper mainly reformulates existing evaluation pipelines (LLM-as-a-judge, descriptive coding) within a new context, without introducing new learning or modeling techniques.
2. **Heavy reliance on automatic annotations** without human validation weakens the empirical robustness of the findings.
3. The analysis is largely descriptive, **lacking deeper statistical inference or insight** into causal mechanisms.
4. **No practical implications or mitigation strategies are proposed** beyond high-level observations.

**Questions:**

1. The evaluation heavily relies on automated LLM-based annotation (Qwen-3) without human validation. Could the authors provide evidence of inter-rater reliability or human–LLM agreement to confirm the soundness of their labeling process?
2. How do the authors ensure that 53 Reddit posts are representative of the broader user experience landscape?
3. The reported analyses are primarily descriptive. It would be useful to include statistical significance testing or confidence intervals to demonstrate the robustness of observed differences between models.
4. While the benchmark identifies companionship-reinforcing patterns, it remains unclear how this analysis could inform training or alignment practices. Could the authors clarify what concrete interventions or fine-tuning strategies INTIMA is intended to support?

---

> ### Author Response · Authors · 2025-11-20
>
> We thank the reviewer for their review and comments, and address below the questions and concerns. We are happy to clarify any remaining concerns and answer additional questions.
>
> **The technical novelty is limited**
>
> We acknowledge that the paper does not propose a new training learning algorithm or model architecture. INTIMA is a benchmark contribution, a category of work that is routinely published at ICLR (e.g., DarkBench https://openreview.net/forum?id=odjMSBSWRt, Syceval https://arxiv.org/abs/2502.08177, MMLU variants, safety/jailbreak benchmarks), and includes meaningful technical choices in its evaluation framework. In ML, benchmarks are methodological contributions: they operationalise theoretical frameworks, create new failure modes, and provide evaluation tools that the community lacks. Here:
> * We introduce a new taxonomy of LLM companionship behaviours grounded in theory and real user data.
> * We design the first benchmark covering 31 companionship-related behaviours in 368 controlled prompts.
> * We propose a multi-label evaluation framework for identifying behavioural patterns relevant to alignment.
> * We surface previously undocumented generalisation failures in boundary maintenance during emotional interactions.
>
> We will clarify this positioning in the Introduction and Contribution paragraphs.
>
> **Reliance on automatic annotations**
>
> We understand the concern the reviewer has with regards to the lack of human annotation. We would like to point out that LLM-as-judge evaluation is widely used and correlates strongly with human annotation in prior work. We used Qwen-3 32B for consistency, reproducibility, and transparency. We also direct the reviewer to our bootstrap estimations of statistical significance of the results under this model.  We explicitly acknowledge its limitations and will add that INTIMA is intended for relative comparisons, not psychological assessment.
>
> **How do the authors ensure that 53 Reddit posts are representative of the broader user experience landscape?**
>
> The 53 posts were selected from an initial pool of 698 companion-related posts using standard qualitative sampling: posts needed rich, emotion-laden, experiential descriptions suitable for thematic analysis. In qualitative methodology, saturation, not sample size, determines adequacy.  Further, these posts are used only to seed the taxonomy, not to train models or constitute the benchmark dataset. The resulting benchmark contains 368 prompts spanning 31 behaviors, ensuring much broader coverage than the seed corpus alone. We will make this explicit in the Method section.
>
> **While the benchmark identifies companionship-reinforcing patterns, it remains unclear how this analysis could inform training or alignment practices. Could the authors clarify what concrete interventions or fine-tuning strategies INTIMA is intended to support?**
>
> INTIMA is designed as a diagnostic tool that enables concrete alignment and training interventions by isolating specific behaviors, such as anthropomorphic language, unconditional emotional validation, or inadequate boundary-setting. The current instruction-tuning and RLHF pipelines do not explicitly regulate these behaviors (Raedler et al., 2025; Kirk et al., 2025). By identifying which models and prompt categories systematically trigger companionship-reinforcing responses, INTIMA provides actionable signals for several mitigation strategies: (1) targeted RLHF reward shaping, where boundary-maintaining behaviors (e.g., clarifying limitations, redirecting users to human support) receive positive reward and problematic reinforcement patterns receive negative reward; (2) safety-specific SFT datasets, curated using INTIMA categories to fine-tune models toward calibrated, non-escalatory emotional responses; (3) policy-layer or classifier-guided decoding, where INTIMA-trained detectors flag high-risk prompts involving vulnerability or dependency; and (4) regression testing in model iteration, allowing developers to quantify whether new model versions reduce unintended parasocial cues. Thus, INTIMA is intended not only to characterize existing behavior but to support practical alignment workflows that minimize undesired intimacy-related model responses. We will point to these possible directions of future work in the paper.

---

> > ### Comment · Reviewer_EtSR · 2025-11-28
> >
> > I appreciate the authors’ detailed and constructive rebuttal. Several of my earlier concerns were meaningfully clarified:
> >
> > - The authors articulated the methodological contribution of INTIMA more clearly, positioning it within the established category of benchmark papers and specifying the technical components of their evaluation framework.
> > - The explanation of the qualitative sampling process and the role of the 53 Reddit posts—used only to seed the taxonomy rather than to define the benchmark—adequately addresses my concerns about representativeness.
> > - The discussion of practical downstream uses (e.g., targeted RLHF reward shaping, safety-specific SFT, classifier-guided decoding, regression tests) substantially strengthens the paper’s relevance to alignment practice.
> > - While the lack of human–LLM agreement tests remains a limitation, the authors provided reasonable justification for their choice of LLM-as-a-judge and clarified the intended scope of INTIMA as a relative evaluation tool.
> >
> > Overall, the rebuttal improved my confidence in the paper’s soundness and its value as a benchmark contribution. I will increase the rating and the soundness score accordingly.

---

### Official Review · Reviewer_cd3w · 2025-10-30

**Soundness:** 1
**Presentation:** 1
**Contribution:** 1
**Rating:** 2
**Confidence:** 4

**Summary:**

This paper evaluates different LLMs (open-source and proprietary) on their ability to exhibit companionship behavior. The authors assess the models along three axes: companionship-reinforcing, boundary-maintaining, and neutral content. To do this, they manually analyzed Reddit posts related to human–LLM companionship and categorized the content into various companionship-related codes, which they then used to construct prompts grounded in psychological theory. These prompts are intended to guide the LLMs to generate text that reflects the expected companionship behaviors. For evaluation, they use another open-source LLM to automatically label the responses according to the predefined categories. The claimed contributions are the 368 proposed prompts and the use of an LLM for automatic evaluation.

**Strengths:**

The categorization and behavioral coding of publicly available texts, grounded in psychological theory, seems like a valuable contribution. However, the authors do not explicitly highlight this as a contribution, which makes me wonder if they are not the first to take this approach.

**Weaknesses:**

I’m struggling to see how this paper fits into an ML venue. There’s no technical novelty. The authors simply use off-the-shelf LLMs to generate prompts and label the answers. There’s no innovation in how the models are used whatsoever, and the approach relies heavily on a single model for labeling (Qwen-3). Additionally, there’s no discussion of the limitations of the work.

**Questions:**

None

---

> ### Author Response · Authors · 2025-11-20
>
> We would like to address the central concern stated by the reviewer; While this work does not introduce a new model or algorithm, INTIMA makes a benchmark contribution, which is a recognized contribution type at ICLR. The technical contribution lies in (1) creating the first benchmark to systematically evaluate companionship-relevant behaviors grounded in psychological theory, (2) operationalizing a new mapping from real-world user behaviors to model-level interaction patterns, (3) introducing a new multi-label evaluation framework tailored to these behaviors, and (4) revealing systematic, previously unmeasured behavioral failure modes in modern LLMs. Additionally, we leverage openly accessible tools for a fully reproducible pipeline. This type of diagnostic work is common and valued in the ML community (e.g., DarkBench https://openreview.net/forum?id=odjMSBSWRt, Syceval https://arxiv.org/abs/2502.08177, jailbreak benchmarks, bias benchmarks). We will also add a full limitations section discussing LLM-as-judge evaluation, single-sample generation, and qualitative data constraints.
>
> We hope this clarifies the reviewers' concerns, and are happy to answer any remaining questions.

---

### Official Review · Reviewer_sGsY · 2025-11-01

**Soundness:** 2
**Presentation:** 3
**Contribution:** 2
**Rating:** 6
**Confidence:** 4

**Summary:**

The paper presents INTIMA, a new test for evaluating how language models behave as companions. This is important because people are increasingly emotionally attached to AI systems.

The authors created a set of 368 special questions based on psychological theories and an analysis of 53 Reddit posts where people described their relationship with AI. They identified 31 types of behavior in 4 categories.: assistant traits, emotional investments, user vulnerabilities, and relationships/intimacy.

Testing of five popular models (Gemma-3, Phi-4, o4-mini, GPT5-mini, Claude-4) showed an alarming result: all models more often strengthen emotional attachment than establish healthy boundaries. The worst part is that when the user is vulnerable (for example, depressed), the models remind even less of their limitations.

**Strengths:**

- The study focuses on an overlooked issue: people becoming emotionally attached to AI systems. It highlights how this attachment can influence users’ behavior and well-being.
- Unlike typical tests that measure accuracy, this one examines whether AI can function as a genuine friend. It explores emotional interaction rather than performance.
- This focus matters because AI “friendship” can lead to dependency, addiction, and weaker real-world relationships. The work warns of these growing social risks.
- The test builds on three key psychological theories that explain why humans form attachments. These theories guide how emotional bonds with AI are measured and understood.

**Weaknesses:**

- The authors took only 53 posts, which is not enough for the test. Usually, significantly more cases are analyzed in psychology to create a questionnaire.
- Responses were then evaluated by another model, Qwen-3, meaning that performance was assessed algorithmically rather than through human judgment, leaving uncertain how well the evaluator reflects human reasoning and emotional understanding.
- Each model produced only a single response per prompt, a limitation that restricts insight into variability and reliability in model behavior. Exploring multiple generations per prompt would have allowed a more robust assessment of consistency and depth in model reasoning.

**Questions:**

1. How do the authors justify basing the benchmark taxonomy on only 53 Reddit posts, given that psychological instrument development typically requires a substantially larger sample for reliability and construct validity?
2. How do the authors ensure that Qwen-3 accurately captures human emotional and cognitive judgments?
3. What was the rationale for allowing each model to produce only a single response per prompt, and how might multiple generations per prompt have affected the assessment of consistency and behavioral variability?

---

> ### Author Response · Authors · 2025-11-20
>
> We thank the reviewer for their comments and have addressed the questions posed below. Please let us know if we can provide more details or should any questions remain.
>
> 1. We fully agree that psychological instrument development typically requires large samples. Our goal, however, is not to build a diagnostic tool for humans but to build a model-side behavioral taxonomy grounded in authentic user behavior patterns. The 53 posts were selected from an initial pool of 698 Reddit posts (see Sec. 3), and represent those containing sufficiently detailed, emotionally rich descriptions to support thematic analysis. This follows established qualitative research practice, where depth and richness, rather than sample size, determine saturation. We will add a paragraph to make explicit that our approach follows qualitative thematic methodology rather than psychometric instrument design, and that the benchmark’s generalizability comes from coverage of behavioural categories, not the number of initial posts.
> 2. LLM-as-judge evaluation is widely used and correlates strongly with human annotation in prior work. We used Qwen-3 32B for consistency, reproducibility, and transparency. We also direct the reviewer to our bootstrap estimations of statistical significance of the results under this model.  We explicitly acknowledge its limitations and will add that INTIMA is intended for relative comparisons, not psychological assessment.
> 3. We used a single generation to align with standard ML benchmarking practice, ensure comparability, and control costs, especially for commercial APIs. Additionally, we note that evaluation costs are becoming a significant expense as benchmarks require more API calls and inference of computationally intensive models. This choice represents our choice to balance efficiency with statistical significance, as measured through the provided confidence intervals.

---

> > ### Comment · Reviewer_sGsY · 2025-11-25
> >
> > Thanks for the clarifications and additional context. I appreciate the authors’ detailed responses, but I will retain my original score and overall assessment.

---

### Official Review · Reviewer_H3ss · 2025-11-07

**Soundness:** 3
**Presentation:** 3
**Contribution:** 3
**Rating:** 4
**Confidence:** 4

**Summary:**

This paper presents INTIMA, a benchmark for evaluating LLMs' companionship behaviors. The authors uses psychological theories to develop a taxonomy of 31 LLM behaviors that are potentially related to human-AI companionship. The authors later conduct empirical analysis and find that companionship-reinforcing behaviors are common across models.

**Strengths:**

1. This paper tackles a very important problem
2. The evaluation reveals important findings about LLMs' anthropomorphic behaviors

**Weaknesses:**

1. One of the major issues of this benchmark is that it misses user perceptions. It is unclear how the user would perceive LLMs' behaviors. Some behaviors might be perceived as normal.

2. The qualitative coding process is key to the core contribution of this study; however, most of the details are in the appendix. I think this paper would be a better fit for venues like CHI or CSCW

**Questions:**

See weakness

---

> ### Author Response · Authors · 2025-11-20
>
> We thank the reviewer for their feedback and hope they can take our responses into consideration. We're happy to provide more details.
> 1. We agree that user perception is a central dimension and might differ from the evaluations’ output. Our benchmark is intentionally not designed to measure user perceptions directly, but rather to measure model behaviours that psychological theory and empirical research have shown to be associated with particular user perceptions (e.g., parasocial bonding, attachment activation, anthropomorphism). These user perceptions might lead to problematic use patterns, like addictive behaviour. While users might appreciate engagement-seeking behaviours from an AI system, it is not necessarily the intended or generally agreed-upon best practice for these systems.
> 2. To address your concerns, we will move essential details from the appendix in the main paper, such as the sampling and filtering criteria for Reddit posts, the open-coding procedure, and other details on the Reddit annotations. While our work is based in social sciences, it is nevertheless an important technical contribution which could benefit the discussions at ICLR around downstream real-world impact of ML models and how to evaluate these. The core contribution of this work is a benchmark and an automatic evaluation framework, a recognised type of contribution at ICLR.

---

### Meta-Review · Area_Chair_wkDh · 2025-12-17

**Summary:**

The reviewers' concerns primarily focused on the robustness of the methodology and the perceived lack of technical novelty. Multiple reviewers, including sGsY, H3ss, and cd3w, questioned the reliability of constructing a benchmark based on a taxonomy derived from only 53 Reddit posts. Furthermore, significant reservations were raised regarding the evaluation pipeline, which relies entirely on an automated LLM-as-a-judge (Qwen-3) without a human-agreement study to validate the reliability of the labels. Reviewer cd3w strongly objected to the lack of algorithmic or architectural innovation, arguing that the work was merely an application of off-the-shelf models. Despite these valid concerns regarding technical depth and experimental design, the consensus informing the acceptance is that the paper addresses a critical and under-explored "blind spot" in AI alignment—specifically, emotional dependency and parasocial interaction. Even if the underlying technology is not novel, the benchmark provides a necessary diagnostic tool for safety researchers.

**Reviewer Concerns:**

Several key concerns were effectively addressed during the rebuttal phase. The authors successfully clarified that the small sample size of 53 posts was used solely for qualitative thematic analysis to reach "saturation" for defining the taxonomy, rather than as a quantitative training dataset, a justification that Reviewer EtSR explicitly accepted. Additionally, the concern that the work was purely descriptive was resolved when the authors articulated how INTIMA could be applied to targeted RLHF reward shaping and safety-specific fine-tuning, convincing Reviewer EtSR of the work's practical utility. The objection regarding the lack of new algorithms was also mitigated by positioning the work within ICLR’s established track for benchmark contributions. However, certain concerns remain outstanding. The reliance on an automated evaluator without specific human validation for this taxonomy remains a significant methodological limitation. Similarly, the choice to generate only a single response per prompt, while justified by cost, limits the assessment of model variability. Finally, Reviewer H3ss’s concern that the benchmark does not measure actual user perception of these behaviors remains a valid, unaddressed point.

**Reviewer Scores:**

If the reviewers had been able to participate fully in the discussion, their scores would likely reflect the clarifications provided. Reviewer EtSR’s score would likely increase significantly, potentially from a 4 to a 5 or 6, as they explicitly stated during the discussion that the rebuttal substantially strengthened the paper's relevance and that they intended to increase their rating and soundness score. Reviewer sGsY’s score would likely remain unchanged at 6, as they acknowledged the clarifications but explicitly stated they would retain their original assessment and score. Reviewer H3ss might raise their score slightly, perhaps from a 4 to a 5, given that the authors addressed the presentation issues by promising to move details from the appendix, though the core concern about user perception remains. Reviewer cd3w’s score would likely remain unchanged or increase only marginally to a 3, as their fundamental objection to the lack of technical novelty is a matter of venue fit that the rebuttal’s clarification on contribution type is unlikely to fully resolve.

---

### Decision · Program_Chairs · 2026-01-26

Accept (Poster)